# Evaluation of Cyclic Fatigue and Bending Resistance of Coronal Preflaring NiTi File Manufactured with Different Heat Treatments

**Soram Oh [1], Su-Young Moon [2], Antonis Chaniotis [3], Eugenio Pedullá [4], Adel Saeed Al-Ghamdi [5], Fawzi Ali Al-Ghamdi [6], Ayman Omar Mandorah [7], Hiran Perinpanayagam [8], Kee-Yeon Kum [9], Ove A. Peters [10,11] and Seok Woo Chang [1,*]**

1. Department of Conservative Dentistry, KyungHee University Dental Hospital, 23 Kyungheedaero, Dongdaemun-gu, Seoul 02447, Korea; soram0123@khu.ac.kr
2. Korea Research Institute of Chemical Technology, 141 Gajeong-ro, Yuseong-gu, Daejeon 34114, Korea; msy1609@krict.re.kr
3. Private Practice, 140 Eleftheriou Venizelou Str., 17676 Kalithea, Greece; antch8@me.com
4. Department of General Surgery and Surgical-Medical Specialties, University of Catania, Via Cervignano, 29, 95129 Catania, SC, Italy; eugeniopedulla@gmail.com
5. King Abdulaziz Hospital, Jeddah 22421, Saudi Arabia; adelll289@hotmail.com
6. Director of Dental Services, Ministry of Health, P.O. Box 109196, Jeddah 21351, Saudi Arabia; Fawzi2003@yahoo.com
7. Department of Restorative and Dental Materials, Faculty of Dentistry, Taif University, Taif 26571, Saudi Arabia; amandorah@tudent.edu.sa
8. Schulich School of Medicine & Dentistry, University of Western Ontario, 1151 Richmond Street, London, ON N6A3K7, Canada; hperinpa@uwo.ca
9. Department of Conservative Dentistry, Dental Research Institute, Seoul National University Dental Hospital, Seoul National University School of Dentistry, 101 Daehak-ro, Jongro-gu, Seoul 03080, Korea; kum6139@snu.ac.kr
10. Departments of Endodontics, University of the Pacific School of Dentistry, San Francisco, CA 94103, USA; opeters@pacific.edu
11. School of Dentistry, University of Queensland, Brisbane, QLD 4000, Australia
* Correspondence: swc2007smc@khu.ac.kr

**Abstract:** Coronal preflaring NiTi files should possess proper hardness, resistance to fracture and flexibility. This study compared the bending resistance and cyclic fatigue resistance of three orifice preflaring NiTi files. HyFlex EDM Orifice opener (#25/12), HyFlex CM (#25/08) and One Flare (#25/09) were tested (n = 46/instrument). Bending resistance was assessed with the stress when perpendicular displacement was applied to a 3 mm point from the file tip, and was performed at either room temperature (RT, n = 10) or body temperature (BT, n = 10). Cyclic fatigue resistance was tested with an artificial ceramic root canal at RT (n = 10) and BT (n = 10), and the number of cycles to failure (NCF) was obtained. The bending resistance and the NCF results were analyzed by two-way ANOVA and Tukey's test. Differential scanning calorimetry (DSC) and X-ray diffraction (XRD) analyses were performed (n = 3, respectively). HyFlex EDM exhibited the highest bending resistance, followed by One Flare and HyFlex CM ($p < 0.05$), irrespective of the tested temperature. At RT, HyFlex CM demonstrated the highest NCF ($p = 0.001$), while HyFlex EDM had the highest NCF at BT ($p < 0.001$). The tested NiTi files were composed of austenite and martensite according to the DSC and XRD results. HyFlex EDM had the highest bending resistance and NCF measured at BT.

**Keywords:** bending resistance; cyclic fatigue; differential scanning calorimetry; orifice opener; X-ray diffraction

## 1. Introduction

Most nickel–titanium (NiTi) rotary file systems recommend a crown-down approach. The enlargement of the coronal region precedes the shaping of the middle and apical region of the root canal. Coronal preflaring reduces the initial coronal curvature of the root canal,

and prevents iatrogenic problems such as ledge, canal blockages or transportation [1,2]. Coronal preflaring facilitates the tactile detection of the apical constriction and increases the accuracy of working length determination [3–5]. The coronal space prepared by a preflaring instrument functions as a reservoir for root canal irrigant to coronally flush the debris resulting from root canal shaping, and reduces the amount of debris extruded apically [6,7]. NiTi rotary files used for coronal preflaring should possess a proper cutting ability and hardness as well as resistance to fracture in the coronal curvature of the root canal.

By applying heat treatment to the manufacturing process, NiTi files have become more flexible and more resistant to cyclic fatigue fracture [8,9]. HyFlex CM (Coltène/Whaledent, Altstätten, Switzerland) and HyFlex EDM (Coltène/Whaledent) are made of controlled memory wire (CM wire; DS Dental, Johnson City, TN, USA). Unlike the conventional superelastic NiTi file, CM wire instruments do not rebound to their original shape when the external load is removed [10]. The HyFlex CM #25/08 instrument, used as a cervical preflaring instrument, has lower bending resistance than conventional Ni-Ti files (ProFile Orifice Shaper #30/06, BioRaCe #25/08, ProTaper universal Sx), while the cutting efficiency of HyFlex CM #25/08 is superior. [11].

HyFlex EDM is manufactured by the electrical discharge machining (EDM) process of CM wire. The EDM process hardens the surface of the NiTi file, resulting in superior resistance to fracture and low degradation is observed after multiple root canal shapings [12]. A number of previous studies have demonstrated the superior resistance to cyclic fatigue fracture of HyFlex EDM compared to a conventional NiTi file, M-wire, gold heat-treated or blue heat-treated NiTi files [13–15]. When compared to HyFlex CM, HyFlex EDM exhibited greater hardness, maximum torque to fracture and cyclic fatigue resistance [12,13,16–18]. The HyFlex EDM orifice opener (#25/12) was released for the preparation of the root canal orifice prior to glide path file, yet little is known about its mechanical properties [19].

OneFlare (MicroMega, Besançon, France) was manufactured through T-wire heat treatment after the grinding process. Ataya et al. [20] compared the mechanical properties of OneFlare with a NiTi file which had identical design but was made with conventional NiTi wire. OneFlare exhibited superior cyclic fatigue resistance and lower bending resistance.

A lot of studies have been conducted on the cutting efficiency and dentinal crack formation of cervical preflaring NiTi files; however, a few studies have assessed flexibility and cyclic fatigue resistance [11,20–23]. Additionally, no study has compared the cyclic fatigue resistance of HyFlex CM #25/08, HyFlex EDM orifice opener, and OneFlare. The objective of this study was to compare the bending resistance and cyclic fatigue resistance of three coronal preflaring NiTi files with different heat treatments.

## 2. Materials and Methods

A total of forty-six instruments of the brands HyFlex EDM Orifice Opener (#25/12), the HyFlex CM with a size of #25/08, and the One Flare (#25/09) were used in the present study; their features are described in Table 1. All instruments were inspected under a dental operating microscope (G3; Global, St. Louis, MO, USA) at 12.8 magnification, no instrument reported signs of deformation and none were discarded. Twenty instruments were subjected to bending resistance performed at either room temperature (RT, n = 10) or body temperature (BT, n = 10), and twenty instruments were tested to cyclic fatigue resistance at RT (n = 10) and BT (n = 10). To evaluate phase transformation and phase composition of the tested files, differential scanning calorimetry (DSC, n = 3) and X-ray diffraction analyses (XRD, n = 3) were performed.

**Table 1.** Specifications of tested files.

| Brand | Size and Taper | Length | Manufacturer |
|---|---|---|---|
| HyFlex CM #25/08 (CM) | #25, 8% taper | 19 mm | Coltène/Whaledent, Altstätten, Switzerland |
| HyFlex EDM orifice opener (EDM) | #25, 12% taper | 15 mm | Coltène/Whaledent, Altstätten, Switzerland |
| One Flare (OF) | #25, 9% taper | 17 mm | MicroMega, Besançon, France |

*2.1. Bending Resistance*

The bending resistance of instruments was evaluated using the cantilever bending test according to previous studies [24,25]. The cantilever bending test measures the load exerted on the cutting blade within elastic displacement, and provides information on the flexibility of the Ni-Ti file [24,26]. The file was inserted into the contra-angle hand-piece of an endodontic motor (X-smart, Dentsply Maillefer, Ballaigues, Switzerland), which was fixed to the lower arm of a universal testing machine (AGS-X STD, Shimadzu, Kyoto, Japan) with vinyl poly-siloxane impression material. The file was placed to be perpendicular to the long axis of the universal testing machine. The file was immersed in distilled water (DW) with a temperature of RT ($22 \pm 0.5$ °C) or BT ($36 \pm 0.5$ °C). A load was applied at a point of 3 mm from the tip of the NiTi file with a stainless steel blade which was connected to the crosshead of the universal testing machine. The load cell was 5000 N, and the crosshead speed was 2 mm/min. Bending resistance was recorded as stress at the point where the blade moved downward 3 mm. The bending resistance of tested files were analyzed by two-way ANOVA (brand of NiTi file vs. experimental temperature) and Tukey's test.

*2.2. Cyclic Fatigue Resistance*

The cyclic fatigue resistance of the instruments was evaluated using a customized device with a static model [27]. Twenty instruments of each brand (EDM, CM, OF) were tested within a curved artificial canal made of lithium disilicate CAD/CAM block (E. max HT A2 shade, Dentsply Sirona), which provided a reproducible curved path to the NiTi files with a 50° angle and 6.4 mm-radius according to Pruetts's method (Figure 1A). The diameter of the artificial root canal was 1.5 mm, the total length was 12 mm, and the center of the curvature was 4.5 mm from the tip of the instrument. Instruments were tested at RT (n = 10) and BT (n = 10). The instrument was mounted to the hand-piece of the endodontic motor (X-smart), which was fixed by a wooden block (Figure 1B,C). The instrument was continuously rotated according to the manufacturers' instructions: 400 rpm for EDM, 500 rpm for CM, and 300 rpm for OF. The time to fracture (seconds) was recorded, then it was multiplied by the operating rpm of the instrument to obtain the number of cycles to failure (NCF). The length of the fractured instrument was measured using a digital caliper (ABS Digimatic caliper CD-10APX; Mitutoyo, Kawasaki, Japan). The NCF data and fracture length of the tested files were analyzed by two-way ANOVA (brand of NiTi file vs. experimental temperature) and Tukey's test. Additionally, fractured files were examined with scanning electron microscopy (SEM, S-4700, Hitachi, Ltd., Tokyo, Japan).

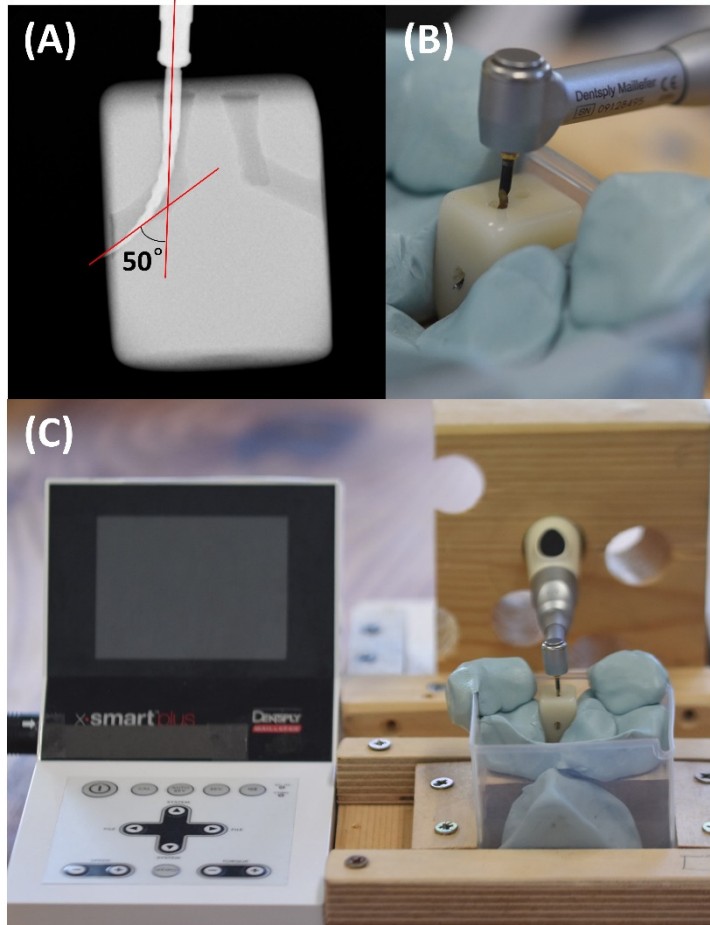

**Figure 1.** Cyclic fatigue resistance test device. (**A**) radiography of artificial root canal; (**B**) ceramic block used for artificial root canal; (**C**) endodontic motor connected to cyclic fatigue testing device.

### 2.3. DSC

Unused instruments of EDM, CM and OF (n = 3) were subjected to DSC (Q1000; TA Instruments, New Castle, DE, USA) to assess phase transformation behavior. The 2–3 mm area from the tip was used, which was operated from −100 °C to 150 °C by use of a liquid nitrogen cooling accessory. The linear heating/cooling rate was 10 °C/min. These heating and cooling cycles were repeated three times for each instrument. Enthalpy changes and onset temperatures for the NiTi phase transformations were obtained with the computer software (Advantage software; TA Instruments , New Castle, DE, USA).

### 2.4. XRD

XRD was performed to identify phases in unused instruments of EDM, CM and OF (n = 3) with an X-ray diffractometer (Rigaku RINT 2000; Rigaku, Tokyo, Japan) in a step size of 0.02. XRD analysis was performed with CuKα irradiation at 40 kV and the tube current of 40 mA at RT; the acquisition range was $35° < 2\theta < 80°$. Peaks were indexed to previous publications for austenite and martensite [28–30].

## 3. Results

### 3.1. Bending Resistance

Both the experimental temperature ($p = 0.017$) and brand of NiTi file ($p < 0.001$) significantly affected the bending resistance, but their interaction was not significant ($p = 0.152$). EDM had the highest bending resistance, followed by OF and CM (Table 2). The bending resistance of CM and EDM were not significantly different between RT and BT, whereas the

bending resistance of OF measured at BT was significantly higher than the one measured at RT ($p = 0.027$).

**Table 2.** The bending resistance, number of cycles to failure (NCF) and fracture length of HyFlex CM (CM), HyFlex EDM (EDM), and OneFlare (OF).

| | | **HyFlex CM** | **HyFlex EDM** | **OneFlare** |
|---|---|---|---|---|
| Bending resistance (N) | RT | 1.799 (0.095) [a] | 4.586 (0.384) [b] | 3.967 (0.197) [c,*] |
| | BT | 1.784 (0.151) [a] | 4.854 (0.493) [b] | 4.371 (0.493) [c,*] |
| NCF | RT | 2410 (684.7) [a,*] | 1508 (447.5) [b] | 899 (176.5) [c,*] |
| | BT | 670 (266.2) [a,*] | 1372.7 (446.5) [b] | 522.5 (193.1) [a,*] |
| Fracture length (mm) | RT | 4.61 (0.51) [a] | 4.24 (0.33) [a,b] | 4.07 (0.17) [b] |
| | BT | 4.78 (0.51) [a] | 4.22 (0.21) [b] | 4.0 (0.24) [b] |

Different lowercase letters in the same row indicate significant difference among NiTi files ($p < 0.05$). * indicates significant difference according to experimental temperature within each NiTi file ($p < 0.05$).

### 3.2. Cyclic Fatigue Resistance

Both the experimental temperature ($p < 0.001$) and brand of NiTi-file ($p < 0.001$) significantly affected the NCF, and their interaction was significant ($p < 0.001$). The NCF and fracture length data are described in Table 2. When the cyclic fatigue resistance test was performed at RT, CM exhibited a higher NCF than EDM ($p = 0.001$), and EDM had a higher NCF than OF ($p = 0.023$), whereas when the test was subjected at BT, EDM had a much higher NCF than those of CM and OF ($p < 0.001$), and those two did not show a significant difference each other ($p = 0.565$). The NCF of EDM did not significantly alter according to environmental temperature, while the NCF of CM and OF was lower at BT than RT ($p < 0.001$ and $p = 0.0002$, respectively). The fracture length of the tested files were not affected by the experimental temperature. The fracture length of CM was longer than those of EDM ($p = 0.002$) and OF ($p < 0.001$) at BT; the fracture length of CM was longer than that of OF ($p = 0.003$) at RT.

The fractured surface showed a typical pattern of brittle fracture. All fractured instruments exhibited several crack points at the edges of the fractured surfaces, and multiple fatigue striations were observed near them (Figure 2). Magnified views of box areas in Figure 2A–C,G–I are presented in Figure 2D–F,J–L, respectively. Multiple fatigue striations, which represent brittle fracture, are observed. Some of them (Figure 2D–F,J) appear to have been fractured several thin layers, where multiple cracks might initiate the fracture. Dimple areas, in a characteristic ductile fracture pattern, were examined in all fractured instruments, marked with a yellow boundary in Figure 2A–C,G–I. EDM had a particularly large area which characterized the ductile fracture (Figure 2B,H) compared to other instruments. Each brand of instrument had a similar fractographic pattern irrespective of the experimental temperature.

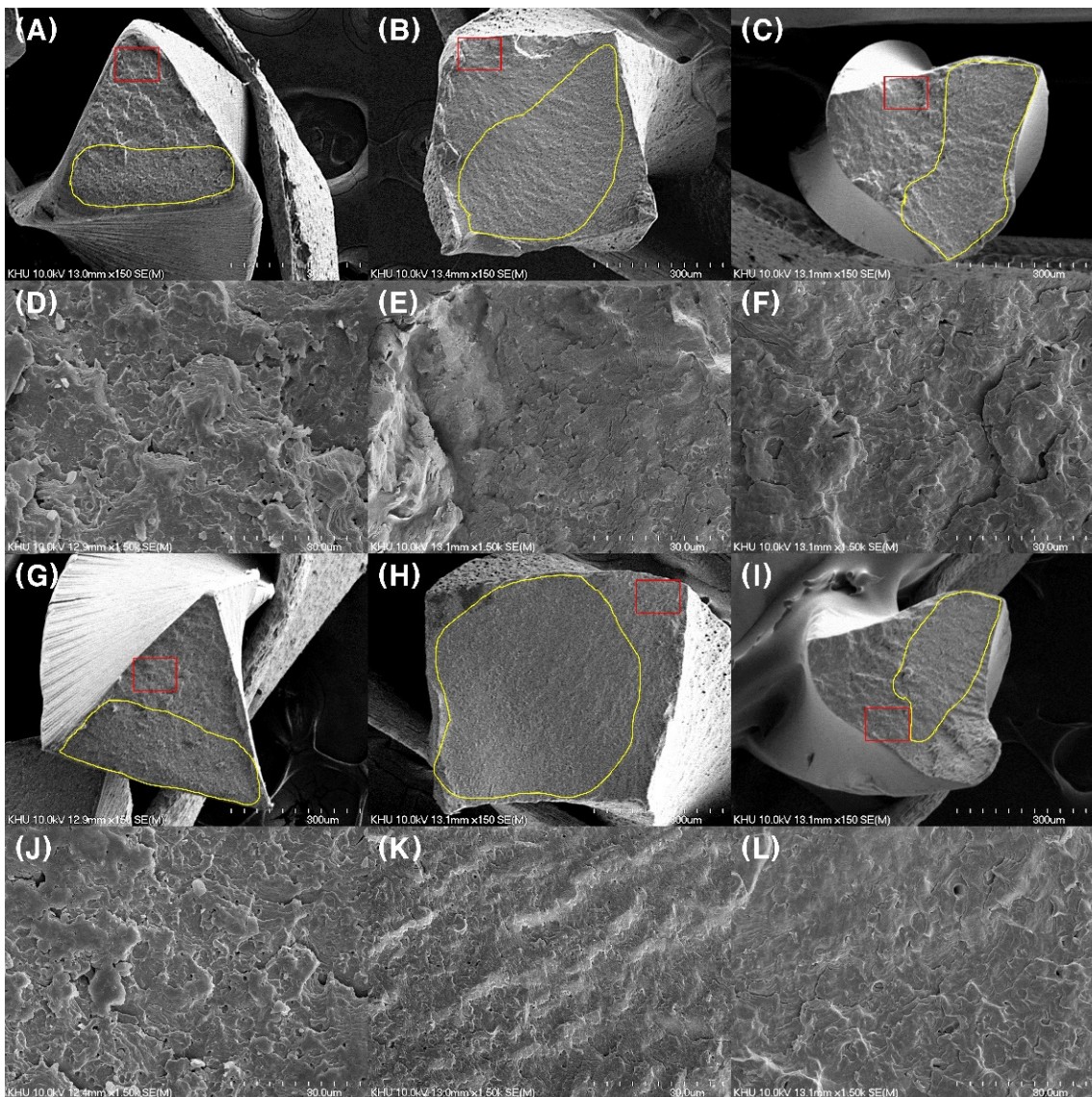

**Figure 2.** SEM images of fractured surface after cyclic fatigue resistance test. (**A–C**) Fractured surface of HyFlex CM, HyFlex EDM, and OneFlare after test performed at RT, ductile fracture area was marked with yellow boundary; (**D–F**) magnified images of box area of (**A–C**), respectively; (**G–I**) fractured surface of HyFlex CM, HyFlex EDM, and OneFlare after test performed at BT, ductile fracture area was marked with yellow boundary; (**J–L**) magnified images of box area of (**G–I**), respectively.

### 3.3. DSC

Figure 3 presents the representative heat flow plots for the CM, EDM, and OF. One definite endothermic peak occurred during the heating cycle (lower curve) of CM (Figure 3A) and OF (Figure 3C), and two indistinct exothermic peaks occurred during the cooling cycle (upper curve), whereas the DSC result of EDM showed one definite endothermic peak as well as two definite exothermic peaks (Figure 3B). Austenite start temperatures of CM, EDM and OF were 27.53, 33.91, and $-8.8\ ^\circ$C, respectively, and austenite finish temperatures of CM, EDM and OF were 58.3, 52.36, and 39.2 $^\circ$C, respectively.

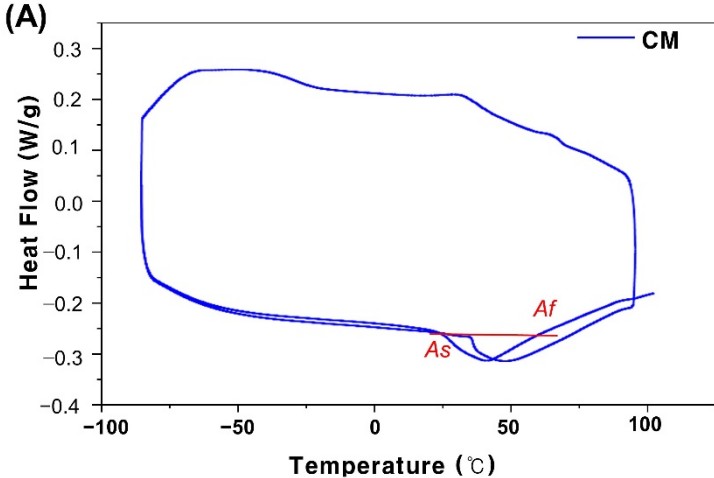

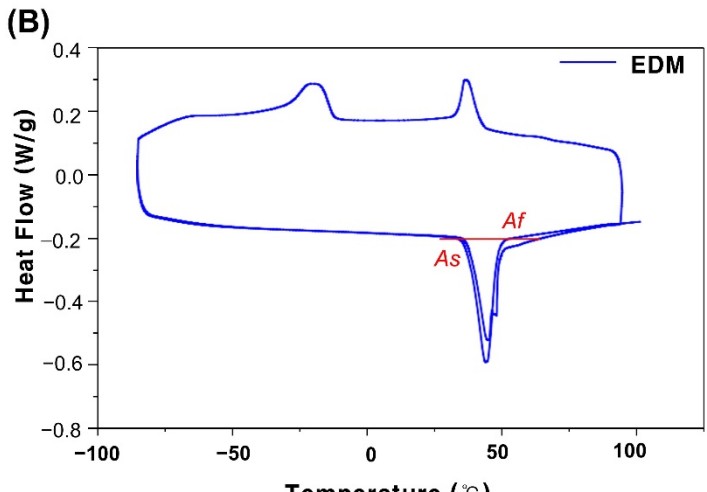

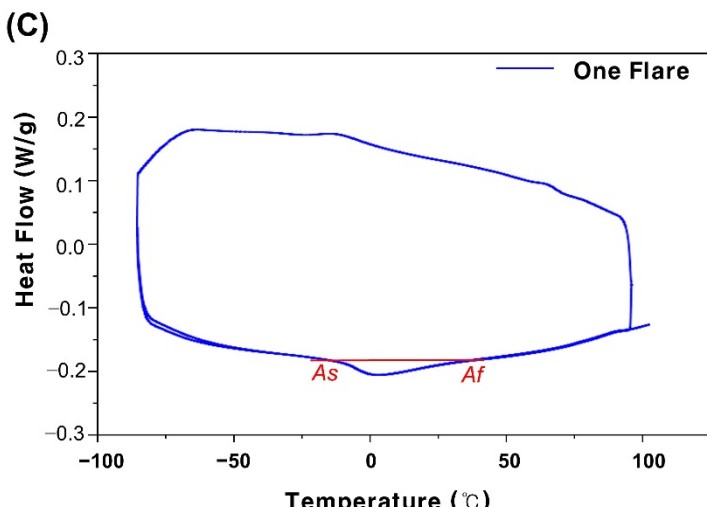

**Figure 3.** Representative DSC results of HyFlex CM (**A**), HyFlex EDM (**B**), and OneFlare (**C**).

The lower curve is the heating cycle and the upper curve is the cooling cycle. The austenite start temperature (As) and austenite finish temperature (Af) are shown in the heating curve.

### 3.4. XRD

XRD patterns contained a major peak for the atomic planes in austenite. A combined peak of austenite and R-phase and a peak for martensite were also identified (Figure 4).

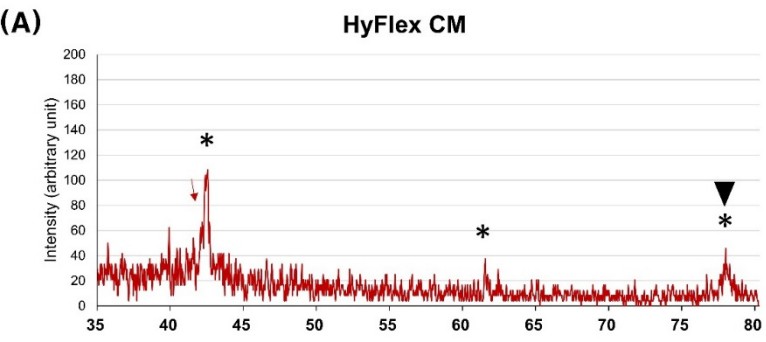

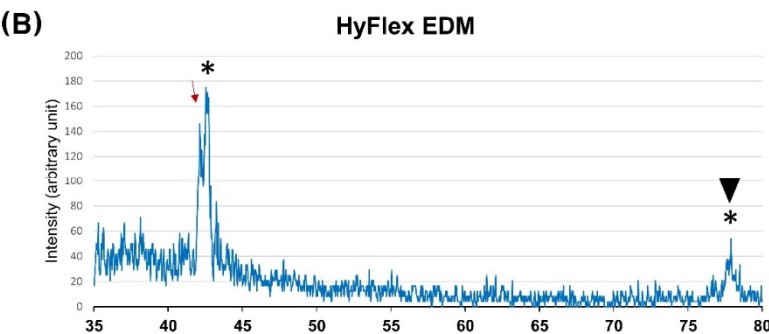

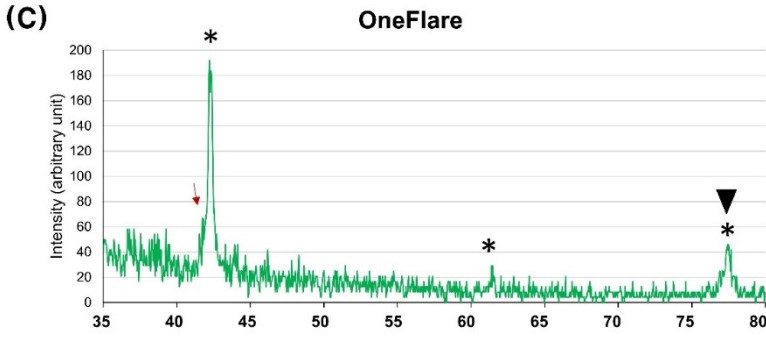

**Figure 4.** XRD results of HyFlex CM (**A**), HyFlex EDM (**B**), and OneFlare (**C**). Peaks for austenite (*), martensite (arrow) and R-phase (arrowhead) are matched.

## 4. Discussion

This study evaluated the bending resistance and cyclic fatigue resistance of three kinds of orifice preflaring NiTi files manufactured via different heat treatments. In the present study, EDM presented the highest bending resistance at both experimental temperatures, and the highest NCF at BT. The cyclic fatigue resistance of the HyFlex EDM Orifice Opener has not been yet investigated. Nonetheless, the outstanding superior cyclic fatigue resistance of HyFlex EDM instruments has been proven in previous studies [13–15,31], which is attributed to its unique manufacturing process. EDM is a noncontact machining procedure which uses a spark that occurs when a discharge occurs between two electrodes [12,15]. EDM does not require direct contact between workpieces; therefore, it prevents mechanical stress such as machining grooves and milling marks produced in the traditional grinding process [10,31].

The higher bending resistance of EDM was due to the bigger dimensions at 3 mm away from tip. The taper of CM, EDM and OF was 8%, 12% and 9%, respectively. When the bending resistance was tested, the vertical blade was positioned at the same position, i.e., on the 3 mm point from the tip. The cross-sectional area at the 3 mm point was the largest in EDM, which resulted in a higher bending resistance of EDM. Besides, the higher hardness of EDM and lack of machining flaws resulted in higher bending resistance [16]. Goo et al. reported the higher bending stiffness of HyFlex EDM compared with HyFlex CM, both #25/06 size [13].

The NCF of EDM was not significantly affected by the experimental temperature, while the NCFs at BT of CM and OF were significantly lower than those measured at RT. This is in agreement with Vasconcelos et al. [32], who reported that the NCF of four different NiTi files were reduced, including Hyflex CM when the cyclic fatigue was tested in water with BT. The reduced NCF at BT could be due to the austenite finish temperature of the tested instruments being higher than the BT. However, when martensite transforms into austenite, the enthalpy change of CM and OF was much lower compared with EDM (Figure 3). It is speculated that the phase transition of martensite to austenite occurs more easily and the austenite phase will be the primary state at BT. According to the XRD results CM, EDM, and OF contained a martensite and R-phase as well as an austenite phase (Figure 4). The three instruments tested were manufactured with different heat treatments; the R-phase was generated during the proprietary heat treatments [10]. EDM exhibited two distinct peaks in the cooling cycle (Figure 3B): the first one is the R-phase and the second one is the martensite peak. CM and OF had two indistinct peaks in the cooling curve (Figure 3A,C), and the presence of the R-phase was supported by the XRD results. Major peaks for austenite were quite similar among three instruments; however, the peak for martensite was lower for OF compared to CM and EDM (Figure 4).

Previous studies reported that cyclic fatigue resistance is reduced by a larger cross-sectional area [33,34]. Since EDM had a 12% taper and CM had an 8% taper, the cross-sectional area was smaller for CM. The smaller cross-sectional area of CM contributed to superior cyclic fatigue resistance compared to EDM at RT. Nevertheless, EDM had a higher NCF than CM at BT, and the alloy properties of the NiTi file seem to have a more decisive effect on cyclic fatigue resistance than the cross-sectional area.

According to the manufacturer, this controlled memory effect helps clinician reduce the preparation errors. However, according to Bürklein et al. [35] there was no significant difference in the straightening of the curved root canal between HyFlex CM and Mtwo (VDW GmbH, Munich, Germany), a conventional superelastic NiTi file. Saber et al. [36] reported that the apical transportation and canal straightening was comparable between HyFlex CM and iRaCe (FKG Dentaire SA, La Chaux-de-Fonds, Switzerland), a conventional superelastic NiTi file, and both files performed efficiently and kept the original canal curvature well. Further studies about the influence of heat-treated orifice preflaring NiTi files on the torque generation to dentin is required.

When NiTi files cut dentin, proper hardness and stiffness are essential. During preflaring preparation, the files are subjected to lateral force. A file which withstands more lateral force would be advantageous. Ataya et al. [20] argued that the lower stiffness and higher cyclic fatigue resistance of OF would be advantageous in the preparation of coronal root dentin with curvature. Nevertheless, the coronal curvature of the root canal could be overcome by tilting the hand-piece as far as possible in the direction of the curvature. According to Pedullà et al. [37], an inclined insertion enhanced the cutting ability of three different glide path NiTi files due to the increased contact surface area between the file and the substrate. Peters et al. [11] reported that a martensitic coronal flaring file removed more bovine dentin compared to austenitic coronal flaring files. In the present study, HyFlex EDM Orifice Opener and HyFlex CM #25/.08 were more martensitic compared to OneFlare. Further studies are needed regarding the cutting efficiency of the tested files, and the relationship between the cutting efficiency and mechanical properties.

The limitation of this study is the comparison of three NiTi files with different cross-sectional designs, instrument lengths, tapers and heat treatments. If the design, instrument length and taper are all identical and heat-treatment is the only variable factor, it will be easier to determine whether the bending the cyclic fatigue resistance are determined by the heat treatment. The effect of bending resistance and cyclic fatigue resistance on the clinical performance of orifice preflaring NiTi files, such as the cutting efficiency and torque generation, need to be investigated.

## 5. Conclusions

EDM Orifice Opener presented the highest bending resistance and cyclic fatigue resistance at BT. Orifice preflaring files manufactured with heat treatments had mixed-phase compositions of austenite, martensite and R-phase.

**Author Contributions:** Conceptualization, S.O. and S.W.C.; Methodology, S.-Y.M., and F.A.A.-G.; Software, H.P., and K.-Y.K.; Validation, A.O.M. and S.W.C.; Formal Analysis A.C., E.P. and O.A.P.; Resources, K.-Y.K. and S.W.C.; Writing—Original Draft Preparation, S.O. and S.W.C.; Writing—Review and Editing, S.-Y.M., A.C., E.P., A.S.A.-G., F.A.A.-G., A.O.M., H.P., K.-Y.K. and O.A.P.; Visualization, A.S.A.-G. and H.P.; Supervision, S.W.C.; Project Administration, S.O. and S.-Y.M. All authors have read and agreed to the published version of the manuscript.

**Funding:** No funding.

**Institutional Review Board Statement:** Not applicable.

**Informed Consent Statement:** Not applicable.

**Data Availability Statement:** Data available on request to corresponding author.

**Conflicts of Interest:** The authors have no conflict of interest.

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
