# Peer review of "Evaluation of Cyclic Fatigue and Bending Resistance of Coronal Preflaring NiTi File Manufactured with Different Heat Treatments"

_applsci, doi:10.3390/app11167694_

Round 1
Reviewer 1 Report
This manuscript has well defined question and research design. It's interesting for readers and the conclusions are supported by the results.
Article is written in an appropriate way but some dana could be presented more appropriately.
Introduction has some part that could be more appropriate for discussion („However, according to Burklein et al…“)
Material and Methods
Missing data on manufacturer information for 'dental operating microscope'.
Missing data about magnification for 'dental operating microscope' inspection of the instruments.
It would be recommended to describe a 'cantilever bending test' application and purpose.
It is not stated whether endodontic motor was fixed with some material to lower arm of universal testing machine.
In what vessel/container was a distilled water?
2.2.Cyclic fatique resistance
I suggest attaching a picture of a 'customized device with a static model'
What was the radius of artificial canal?
Missing data on manufacturer information for 'digital caliper'
Discussion
I suggest to explain the possible influence of the taper on the results with regard to HyFlex EDM orifice opener (12%) and HyFlex CM (8%)
English level is appropriate and understandable.
Manuscript should be accept after minor revision.
Author Response
This manuscript has well defined question and research design. It's interesting for readers and the conclusions are supported by the results.
Article is written in an appropriate way but some dana could be presented more appropriately.
point 1: Introduction has some part that could be more appropriate for discussion („However, according to Burklein et al…“)’
Response 1: Thank you for your valuable comments. That part moved to discussion section in the revised manuscript.
Material and Methods
Point2: Missing data on manufacturer information for 'dental operating microscope'.
Missing data about magnification for 'dental operating microscope' inspection of the instruments.
Response 2: Thank you for your valuable comments. The manufacturer and magnification of dental operating microscope was added in the revised manuscript as follows.
All instruments were inspected under dental operating microscope (G3; Global, St. Louis, MO, USA) at 12.8 magnification,
Point 3: It would be recommended to describe a 'cantilever bending test' application and purpose.
Response 3: Thank you for your valuable comments. The application and purpose of cantilever bending test was inserted in the revised manuscript as follows.
The cantilever bending test measures the load exerted on the cutting blade within elastic displacement, and provides information on the flexibility of the Ni-Ti file.
Point 4: It is not stated whether endodontic motor was fixed with some material to lower arm of universal testing machine.
Response 4: Thank you for your valuable comments. The hand-piece of endodontic motor was fixed to lower arm of universal testing machine with viny poly-siloxane impression material. That was inserted in the revised manuscript.
Point 5: In what vessel/container was a distilled water?
Response 5: Thank you for your valuable comments. For both bending resistance test and cyclic fatigue resistance test, the instrument was tested by immersing in distilled water in a plastic container. Distilled water was replaced every time the instrument was exchanged to keep the temperature constant. And heater was used in the laboratory to prevent a sharp decline in ambient temperature. For the bending resistance test, the experimental time took 90 to 100 seconds per instrument. For the cyclic fatigue resistance test, the time before fracture ranged from 56 to 431 seconds. The temperature was checked by putting a thermometer in distilled water during the cyclic fatigue resistance test. If the temperature drops by more than 1℃, the water was partially removed with a 10-ml syringe and heated water was supplied to keep the temperature constant.
2.2.Cyclic fatique resistance
Point 6: I suggest attaching a picture of a 'customized device with a static model'
Response 6: Thank you for your valuable comments. The picture of testing device of cyclic fatigue resistance was inserted in the revised manuscript.
Point 7: What was the radius of artificial canal?
Response 7: Thank you for your valuable comments. The radius and angle of curvature was added according to Pruett’s method in the revised manuscript as follows.
with 50° angle and 6.4mm-radius according to Pruetts’s method (Fig. 1A). The diameter of the artificial root canal was 1.5mm, total length was 12mm, and the center of the curvature was 4.5mm from the tip of the instrument.
Point 8: Missing data on manufacturer information for 'digital caliper'
Response 8: Thank you for your valuable comments. The manufacturer and the city, country was inserted in the revised manuscript as follows.
The length of fractured instrument was measured using a digital caliper (ABS Digimatic caliper CD-10APX; Mitutoyo, Kawasaki, Japan).
Discussion
Point 9: I suggest to explain the possible influence of the taper on the results with regard to HyFlex EDM orifice opener (12%) and HyFlex CM (8%)
Response 9: Thank you for your valuable comments. The influence of the taper on the bending resistance test was revised in the discussion section. And the influence of the taper on the cyclic fatigue resistance was inserted in the revised manuscript as follows.
The cross-sectional area at 3mm point was the largest in EDM, which resulted in higher bending resistance of EDM.
Previous studies reported that cyclic fatigue resistance is reduced by larger cross-sectional area. Since EDM had 12% taper and CM had 8% taper, the cross-sectional area is smaller for CM. The smaller cross-sectional area of CM contributed to superior cyclic fatigue resistance than EDM at RT. Nevertheless, EDM had higher NCF than CM at BT, the alloy properties of the NiTi file seem to have a more decisive effect of cyclic fatigue resistance than cross-sectional area.
English level is appropriate and understandable.
Manuscript should be accept after minor revision.
Reviewer 2 Report
This research is under the scope of this journal; the topic is relevant for readers, and this research deals with potentially significant knowledge of the field.
However, there are some concerns about the present manuscript:
Abstract
- In the results, is important to show more information, add some of the p-values.
Introduction
- (Statement of Relevance)
- What is the importance of this study for clinical? What is the gap in this field of literature?
- You do not think this study is included in the others already done? Which results are comparable with other studies? What has this study been new?
- Page 2 lines 52-71 - “When compared to HyFlex CM, HyFlex EDM exhibited greater hardness, maximum torque to fracture and cyclic fatigue resistance” add a reference for cycle fatigue study using a dynamic movement of the files, https://doi.org/10.1007/s10266-018-0401-2. And this for static model: doi: 10.1016/j.joen.2017.05.023 or https://doi.org/10.1016/j.rpemd.2015.11.007 and doi: 10.1111/iej.12470
Materials and Methods
- How was the sample calculated? Did the authors perform a power analysis to evaluate if this sample size was appropriate?
Results
- Please add more information on SEM images, in Figures 1D-F and 1J-L and identified in the manuscript the images. What did you want to show? Please, clarified this point!
Discussion
- Please, clarified other limitations of this study?
- And, clarified the future perspectives.
References
- Please, correct some typos in the references.
- References are not standardized. It has the reference numbers indexed twice.
Author Response
This research is under the scope of this journal; the topic is relevant for readers, and this research deals with potentially significant knowledge of the field. However, there are some concerns about the present manuscript: Abstract
• Point 1: In the results, is important to show more information, add some of the p-values.
Response 1: Thank you for the great comments. I reinforced the results in the abstract section as follows. The limit of the number of words in the abstract is 200, the revised abstract has 197 words. HyFlex EDM exhibited the highest bending resistance, followed by One Flare and HyFlex CM (p<0.05), irrespective of tested temperature. At RT HyFlex CM demonstrated the highest NCF (p=0.001), while HyFlex EDM had the highest NCF at BT (p<0.001). The tested NiTi files were composed of austenite and martensite according to DSC and XRD results. Introduction • (Statement of Relevance)
• Point 2: What is the importance of this study for clinical? What is the gap in this field of literature?
Response 2: Thank you for the great comments. Introduction part was revised, the gap in this filed of literature was inserted as follows. HyFlex EDM orifice opener (#25/12) was released for preparing root canal orifice prior to glide path file, yet little is known about its mechanical properties. A lot of studies have been conducted on the cutting efficiency and dentinal crack formation of cervical preflaring NiTi files, however, a few studies have assessed flexibility and cyclic fatigue resistance
• Point 3: You do not think this study is included in the others already done? Which results are comparable with other studies? What has this study been new?
Response 3: Thank you for the great comments. The introduction part was revised as follows. A lot of studies have been conducted on the cutting efficiency and dentinal crack formation of cervical preflaring NiTi files, however, a few studies have assessed flexibility and cyclic fatigue resistance [11, 20-23]. And there have been no study to compare the cyclic fatigue resistance of HyFlex CM #25/08, HyFlex EDM orifice opener, and OneFlare.
• Point 4: Page 2 lines 52-71 - “When compared to HyFlex CM, HyFlex EDM exhibited greater hardness, maximum torque to fracture and cyclic fatigue resistance” add a reference for cycle fatigue study using a dynamic movement of the files, https://doi.org/10.1007/s10266-018-0401-2. And this for static model: doi: 10.1016/j.joen.2017.05.023 or https://doi.org/10.1016/j.rpemd.2015.11.007 and doi: 10.1111/iej.12470
Response 4: Thank you for your valuable comments. The references were added in the revised manuscript. Materials and Methods
Point 5: - How was the sample calculated? Did the authors perform a power analysis to evaluate if this sample size was appropriate?
Response 5: Thank you for the great comments. We did not perform a power analysis. Instead, we followed previous studies which compared cyclic fatigue resistance of different NiTi files, such as Keles A et al. Effect of temperature of sodium hypochlorite on cyclic fatigue resistance of heat-treated reciprocating files. Journal of endodontics 2019; 45(2): 205-208. (DOI: 10.1016/j.joen.2018.11.003), and Klymus ME al. Effect of temperature on the cyclic fatigue resistance of thermally treated reciprocating instruments. Clinical oral investigations. 2019; 23(7): 3047-3052. (DOI: 10.1007/s00784-018-2718-1) Results
• Point 6: Please add more information on SEM images, in Figures 1D-F and 1J-L and identified in the manuscript the images. What did you want to show? Please, clarified this point!
Response 6: Thank you for the great comments. Detailed description was added in the revised manuscript as follows. Fractured surface showed typical pattern of brittle fracture. All fractured instruments exhibited several crack points at the edges of the fractured surfaces, and multiple fatigue striations were observed near them (Fig. 2). Magnified views of box areas in Fig. 2A-C and Fig. 2G-I are presented in Fig. 2 D-F, and Fig. 2 J-L, respectively. Multiple fatigue striations, which represent brittle fracture, are observed. Some of them (Fig. 2 D-F, J) appears to have been fractured several thin layers, where multiple cracks might initiate the fracture. Dimple areas, characteristic ductile fracture pattern, were examined in all fractured instruments, marked with yellow boundary in figure 2 (A-C, G-I). EDM had a particularly large area which characterized the ductile fracture (Fig. 2B, 2H) compared to other instruments. Each brand of instrument had similar fractographic pattern irrespective of experimental temperature. Discussion
• Point 7: Please, clarified other limitations of this study?
Response 7: Thank you for the great comments. I inserted the limitation of this study in the end of discussion section as follows. The limitation of this study is to compare three NiTi files with different cross-sectional design, instrument length, taper and heat treatment. If the design, instrument length and taper are all identical and heat-treatment is the only variable factor, it will be more clear whether the bending, cyclic fatigue resistance were determined by heat-treatment.
• Point 8: And, clarified the future perspectives.
Response 8: Thank you for the great comments. Future perspective was added in the end of discussion section as follows. The effect of bending resistance and cyclic fatigue resistance on clinical performance of orifice preflaring NiTi files, such as cutting efficiency and torque generation are need to be investigated. References
• Point 9: Please, correct some typos in the references.
Response 9: Thank you for your valuable comments. We corrected the typos of references.
• Point 10: References are not standardized. It has the reference numbers indexed twice.
Response 10: Thank you for your valuable comments. We corrected the reference numbers.
Round 2
Reviewer 2 Report
This research is under the scope of this journal; the topic is interesting for readers and this research deals with potentially significant knowledge to the field and an open new way for future studies.
The authors improved the quality of the manuscript after the reviewer's indications.